# Long-Term Results of Surgical Treatment for Popliteal Artery Entrapment Syndrome

**DOI:** 10.3390/diagnostics14121302

**Published:** 2024-06-19

**Authors:** Wassim Mansour, Francesca Miceli, Alessia Di Girolamo, Ada Dajci, Antonio Marzano, Rocco Cangiano, Marta Ascione, Luca di Marzo

**Affiliations:** Vascular and Endovascular Surgery Division, Department of General Surgery and Surgical Specialties, Policlinico Umberto I, “Sapienza” University of Rome, Viale del Policlinico, 155, 00161 Rome, Italy; alessia.digirolamo@uniroma1.it (A.D.G.); ada.dajci@uniroma1.it (A.D.); antonio.marzano@uniroma1.it (A.M.); rocco.cangiano@uniroma1.it (R.C.); marta.ascione@uniroma1.it (M.A.); luca.dimarzo@uniroma1.it (L.d.M.)

**Keywords:** popliteal entrapment syndrome, duplex ultrasound, musculotendinous section, venous bypass

## Abstract

Introduction: Popliteal artery entrapment syndrome (PAES) is a rare disease of the lower limbs, mainly affecting young patients, due to extrinsic compression of the neurovascular bundle at the popliteal fossa. The aim of this study was to describe our experience during a median 15-year period. Methods: Patients treated for PAES in our institution from 1979 to 2024 were included. Preoperative, intraoperative, and postoperative data were analyzed. Results: A total of 47 patients with a total of 78 limbs were treated. Duplex ultrasound with active maneuvers was performed in all limbs (100%). Angiography was performed in almost all patients (97.4%), computed tomography angiography in 56 (71.8%), and magnetic resonance angiography in 22 (28.2%). Concerning surgical treatment, musculotendinous section was performed in 60 limbs (76.9%), and autologous venous bypass was achieved in 18 limbs (23.1%). The rates for freedom from target lesion revascularization—meaning that no significant stenosis or occlusion during follow-up required revascularization—and 15-year primary patency were 92.4% and 98%, respectively. Conclusion: Long-term results of surgical treatment for PAES seem to be very satisfying. Myotomy with or without arterial reconstruction using venous bypass can lead to good patency at 15 years of follow-up.

## 1. Introduction

Popliteal entrapment syndrome (PES) is an uncommon disease of the lower limbs, mainly affecting athletes or military personnel and young patients without any atherosclerotic risk factors. It results mainly due to compression of the neurovascular bundle at the popliteal fossa by extrinsic popliteal musculotendinous structures. The syndrome can be classified as either anatomical or functional, and six different types are described according to the PVE Forum classification (Table 1) [1].

Anatomical PES is caused by vascular compression because of anatomic anomalies that are frequently present from birth and develop over time. In functional PES, on the other hand, the compression is caused by dynamic factors, such as contraction of the hypertrophied muscles during physical activity.

The type of popliteal structure involved determines the different clinical scenarios. Venous compression is primarily responsible for calf swelling, discoloration, and paresthesia. Meanwhile, arterial compression is primarily responsible for exercise intolerance; lifestyle-limiting claudication; less commonly, post-stenotic aneurysm formation; and, in worst cases, limb-threatening ischemia.

Our study focuses on popliteal artery entrapment syndrome (PAES), in which, in most cases, the vascular injury or mechanical compression requires surgical correction due to aberrant interaction between the popliteal artery and the muscular-tendon structures in the popliteal fossa.

Therefore, the aim of this study was to describe our institutional experience in terms of diagnosis, surgical management, and long-term results of PAES during a 15-year follow-up period.

### 1.1. Pathophysiology

The popliteal neurovascular bundle normally passes between the medial and lateral heads of the gastrocnemius muscle. Regarding PAES, its pathogenic process has an embryological basis related to the development of the popliteal artery and the surrounding musculature.

The proximal part of the popliteal artery originates as a continuation of the superficial femoral artery, which originates from the fusion of the ischiatic artery and the femoral artery. Meanwhile, the union of the tibialis anterior and tibioperoneal trunk forms the distal part, which occurs prior to the migration of the head portion of the medial part of the gastrocnemius into the medial location of the popliteal fossa.

The most encountered variation of a muscle in PAES is that of the medial head of the gastrocnemius muscle. In humans, the muscle mass that is to make up the future medial head of the gastrocnemius muscle migrates across the popliteal fossa from its original lateral position.

The early formation of the distal portion of the popliteal artery and the delayed migration of the proximal head of the medial gastrocnemius muscle cause the first and second types of compressive mechanisms [2].

In the absence of anatomical abnormalities, hypertrophy of the medial and lateral portions of the gastrocnemius muscles intermittently causes compression of the popliteal artery during plantar flexion, resulting in the sixth type of this syndrome, known as the functional subtype.

### 1.2. Incidence

The exact incidence of PES is not clear. Clinical studies report an incidence of 0.17%, while post-mortem studies suggest an incidence of 3.5% [1,2,3]. In a similar study performed by the Popliteal Vascular Entrapment Forum, an incidence of 4.3% was recorded on 162 limbs [3].

Most patients affected by PES are predominantly young (in their early 30 s) males (83%) [4]. Hypertrophy of the medial gastrocnemius muscle observed in athletes has been widely linked to functional PAES, with approximately 60% of reported cases affecting young athletes under 30 years [5].

Nevertheless, most functional PAES are not diagnosed. Nowadays, the diagnosis remains challenging because dynamic imaging is needed as the imaging examinations at rest may be normal [6].

### 1.3. Signs and Symptoms

The clinical characteristics of anatomical PAES vary according to type. A grading classification of the clinical severity of popliteal artery entrapment syndrome has been proposed.

PAES may be asymptomatic or may present with intermittent claudication. In rare cases, it can also present with cold feet and the absence of a distal pulse up, with acute limb-threatening ischemia being the most serious and critical manifestation.

Repetitive trauma to the popliteal artery from focal impingement results in chronic inflammation, followed by occlusive or aneurysmal formation. However, this type is underestimated because only dynamic maneuvers can detect it. Nevertheless, to avoid the disease progression and associated critical complications, surgeons and physicians should not underestimate a clinical suspicion of such pathology [7].

### 1.4. Diagnosis

Nowadays, a specific diagnostic protocol for suspected PAES has not reached a worldwide consensus. Many studies published in the literature report, as first instance, bedside noninvasive clinical tests, such as the use of the ankle-brachial indices (ABI) and duplex ultrasound [8].

However, these tests should be incorporated with provocative maneuvers. Typically, a decrease in oscillometric deflections is observed when the gastrocnemius muscle is actively contracted by plantar flexion or overstretched by passive dorsiflexion of the foot.

If the initial evaluation is indicative of PAES, cross-sectional images are suggested. Computed tomography angiography (CTA) and magnetic resonance angiography (MRA) delineate the underlying musculotendinous anatomical abnormalities.

MRA is known as the “gold standard” in the identification of abnormal popliteal fossa myofascial anatomy related to PAES Types I–VI because of its great sensitivity [4,9].

However, to confirm the diagnosis, in most cases, a digital subtraction angiography is necessary. Most often, a medial shift of the artery may be observed, or a focal stenosis or, in rare cases, a post-stenotic aneurysm formation. At least but not at last, run-off vessel evaluations are mandatory in those cases in which a reconstruction is needed.

### 1.5. Surgical Treatment

Endovascular treatment is the standard of care for revascularization in patients affected by atherosclerotic peripheral artery disease. Surgery is considered the gold standard for PAES. Surgical management includes the release of any fibrous structures and sections of the muscular structures compressing the artery. Most often, the medial head of the gastrocnemius muscle is involved, and its section is mandatory. Hypertrophy of soleus and plantaris muscles or the lateral head of gastrocnemius requires their resection.

In advanced cases with arterial damage, such as arterial stenosis or occlusion, revascularization surgery is needed.

Percutaneous transluminal angioplasty (PTA) may be achieved. However, avoiding the positioning of any stent in this anatomic district is strongly recommended because stent fracture and occlusion are common. Since most patients are active young people, long-term patency of the revascularization procedures should be warranted. Thus, venous bypass grafting is a valid option.

## 2. Materials and Methods

Data from all patients treated for PES in our institution from 1979 to 2024 were retrospectively collected and prospectively analyzed. Patients treated for venous PES were excluded from this study. Only patients affected by PAES were included. Informed consent was obtained, instead ethical approval was waived due to the retro-spective nature of the study, according to our Ethical Committee.

Preoperative, intraoperative, and postoperative data were extrapolated and inserted in a dedicated database.

The preoperative risk factors considered for analysis were sex, smoke habit, heart function, diabetes, Chronic Obstructive Pulmonary Disease (COPD), Chronic Kidney Disease (CKD), and sports attitude.

Each patient was asked if a previously incorrect diagnosis was suspected or if a previous surgery was performed for the same symptoms.

Patients were classified as class 0 if they were asymptomatic, class 1 if they were affected by paresthesia and cold foot, class 2 if they were affected by intermittent claudication after more than 100 m, class 3 if they were affected by intermittent claudication after less than 100 m, class 4 if they were affected by rest pain, and class 5 if a distal gangrene or necrosis was present.

All patients were preoperatively submitted to Doppler CW, with active maneuvers, such as plantarflexion against resistance, to suspect the entrapment syndrome.

To assess for arterial compression, under duplex ultrasound, a popliteal artery scan is carried out on a supine patient while the lower leg muscles are relaxed and in plantar flexion. The patient then flexes the ankle against resistance, frequently displaying full artery blockage in some individuals.

Patients were then submitted to digital subtraction angiography (DSA) with dynamic maneuvers to confirm the diagnosis and to evaluate the popliteal status and the distal vessel run-off, beyond the positivity on dynamic maneuvers.

The popliteal status was evaluated as normal, altered, occluded, aneurysmatic, or stenotic.

Run-off vessels were classified based on the number of patent vessels, with localized, severe, or diffuse atherosclerosis with occlusion of one to two vessels.

At the last examination, a CTA or an MRA was performed to confirm the presence of abnormal structures compromising the popliteal fossa.

Patients were stratified into six different subtypes, according to our PVE Forum classification.

The surgical approach was chosen between medial and posterior depending on the extension of the arterial pathology and the type of myofascial structures involved. After popliteal fossa neurovascular bundle exposure, the structure compressing the vessel was identified and sectioned.

In case of vascular dilatation or occlusion, the aneurysm was resected, and a bypass grafting was performed. Whenever the distal popliteal artery was not patent, a distal bypass was performed. The main conduit used was the great saphenous vein, in an inverted fashion.

Every patient was enrolled in our dedicated follow-up including physical examination and duplex ultrasound at 6 months, 1 year, and annually thereafter.

Postoperative complication, reintervention, or symptom reappearance was recorded at 30 days, 6 months, and annually thereafter.

The primary endpoint was considered symptom regression, and the secondary outcomes were the vascular conduit primary patency, in case of vascular reconstruction.

### Statistical Analysis

Comparative analyses were executed using the χ^2^ test and Fisher’s exact test, contingent upon the data. IBM SPSS Statistics for Windows v.25 (IBM Corp., Armonk, NY, USA), was the tool of choice for statistical analysis. Continuous variables were articulated as means, while categorical variables were represented as percentages. A *p*-value of ≤0.05 was the threshold for statistical significance.

## 3. Results

### 3.1. Patients’ Characteristics

From 1979 to 2024, a total of 76 patients and 118 limbs affected by PES were identified. A total of 40 limbs were treated for venous PES but were excluded from this study. However, we identified 47 patients affected by popliteal artery entrapment syndrome at our academic institution between 1979 and 2024. A total of 78 limbs were treated for PAES. A total of 31 patients (39.7%) had a history of contralateral PAES. The median age of patients was 34 (range, 14–62), who were mostly male (77.2%). The majority were athletes, with only 6 (12.76%) claiming not to be a sports player.

Regarding the initial symptomatology, 4 cases (5.1%) were asymptomatic, 32 (41%) had symptoms of class 1, 30 (38.5%) of class 2, 10 (12.8%) of class 3, and 2 (2.6%) of class 5 (Table 2).

Of those presenting symptoms of exertional pain, 10 patients received a previous diagnosis of chronic compartment syndrome.

### 3.2. Diagnosis of Popliteal Artery Entrapment Syndrome

Duplex ultrasound with active maneuvers, such as dorsiflexion and plantarflexion, during examination was performed on all limbs (100%). Positive maneuvers were found in 76 limbs (97.4%). DSA was performed in nearly all patients (97.4%), whereas CTA in 56 (71.8%) and MRA in 22 (28.2%) (Table 3).

The sensitivity to the detection of PAES was 96.2% for DSA with dynamic maneuvers, 54.5% (12 limbs) for MRA, and 51.7% (29 limbs) for CTA (Figure 1 and Figure 2). 

Post-stenotic aneurysm detection was found in 8 limbs, popliteal artery occlusion in 6 and 8 popliteal stenosis. Regarding run-off vessels, 58 limbs had 3 distal vessels patents, 15 had 2 vessels, 4 only 1 vessel patent, and 1 no run-off was found.

According to the PVE Forum classification, the limbs were stratified based on radiological and intra-operative findings as types II (31, 39.7%), III (23, 29.5%), and VI (24, 30.8%) (Table 2).

### 3.3. Surgical Treatment

In terms of surgical treatment, all 78 limbs were surgically treated. The preferred approach was the posterior approach (61 limbs, 78.2%), whereas 17 (21.7%) received the medial. Musculotendinous section (MTS) was performed in 60 limbs (76.9%). In the remaining 18 limbs, MTS was not performed, and a release of the musculotendinous structures was performed.

The procedural extent included myotomy of the medial gastrocnemius (31 limbs, 51.6%), lateral gastrocnemius (5 limbs, 8.3%), arteriolysis (4 limbs, 5.12%), accessory head of the medial gastrocnemius (2 limbs, 3.3%), additional third head of gastrocnemius (10 limbs, 16.7%), a soleus sling (6 limbs, 10%), and 3 limbs with a fibrous band section.

MTS associated with PTA was performed in 2 limbs (2.5%) (Table 4).

In 4 limbs, preoperative fibrinolysis with urokinase was performed to restore distal outflow. An autologous venous bypass grafting was achieved in 18 limbs (23.1%), as only 1 limb received polytetrafluoroethylene (PTFE) graft. The preferred autologous vein graft was the great saphenous vein.

### 3.4. Postoperative Results

There were thirteen early postoperative complications (16.6%): 3 dehiscent wounds, 3 hematomas, 3 perimalleolar edema, 2 infectious wounds, and 2 lymphorrhea. However, preoperative symptoms improved in >97.4% of limbs at the first postoperative follow-up.

During the mean long-term postoperative follow-up of 181 months (range, 28–480 months), only 1 patient, who received an autologous venous bypass grafting (although poor run-off vessels were found preoperatively), underwent transmetatarsal amputation after 13 months from reconstruction due to occlusion of the graft. New reconstruction due to occlusion of venous bypass grafting was performed in 2 limbs, 1 after 7 months from the first intervention and 1 after 21 months. At a mean follow-up of 181 months, the rates for freedom from target lesion revascularization (TLR) and 15-year primary patency for the surgical treatment of PAES were 92.4% and 98%, respectively.

Those patients treated with MTS had a 15-year patency rate of 98%, whereas those treated with reconstruction had a 71% of patency rate at a 15-year follow-up (*p* < 0.001).

## 4. Discussion

This study reports one of the largest series of PAES with the longest follow-up reported in the literature, to the best of the authors’ knowledge. Since the incidence of this syndrome is quite low, our center had the privilege to be a national referral center for PAES with a large series collected in a prospective database [10,11].

As is well known, most patients affected by PAES are young men, and in some cases, are children, as in our experience, there were 5 cases of child patients. The most common symptom of PAES in pediatric age was claudication, but unfortunately, acute limb ischemia (ALI) was also common [12].

The median percentage of ALI was 11% in a recent meta-analysis on PAES that included primarily adult patients [13].

Since PAES is a progressive disorder, popliteal artery microtrauma and compression exerted by muscular or tendinous abnormalities could lead to arterial damage, with thrombosis or aneurysmal degeneration [14].

This process usually has a chronic evolution, giving the possibilities to develop a collateral network, which can compensate in case of complete occlusion.

For the same reason, early diagnosis is essential to prevent limb ischemia and irreversible arterial damage [13,14], and it is critical to receive treatment as soon as possible to avoid major complications and the possibility of amputation. However, it is important to rule out other potential causes of ALI in pediatric patients, including premature accelerated atherosclerosis, microemboli, Takayasu’s arteritis, collagen vascular disease, coagulopathy, and the presence of a cystic adventitial disease (CAD). Notably, CAD is a rare condition, accounting for about 0.1% of vascular diseases [15]. It is characterized by the development of a cystic mass in the subadventitial layer of the vessel. The pathognomonic sign of this condition is the visualization of a thin, echogenic line separating the lumen of the vessel and the cyst, with the narrowed lumen presenting with an ultrasonic scimitar sign on DUS imaging. In most cases, the neurovascular evaluation is normal. Passive flexion of the knee, however, can result in reduced distal pulses. This differs from PAES, where active plantar flexion or passive dorsiflexion maneuvers contracting the gastrocnemius muscle always reduce pedal pulses. Consequently, it is possible that a physical examination and diagnostic tests will not be able to differentiate between CAD and PAES; therefore, surgical planning should account for the possibility of encountering either condition.

Settembre et al. [13] report a literature review about PAES in children, with 18 cases of ALI, and complications occurred in four children, but full recovery was obtained in all cases, and no major amputations were registered. In our experience, 5 patients who were in their childhood were all successfully treated, without sequelae and with complete symptom regression.

PAES is frequently under-reported or misdiagnosed. There is some evidence in the literature reporting an average of 2 years before receiving a PAES diagnosis; others experienced symptoms for almost 10 years before the cause was identified.

In our series, 10 patients with exertional pain received a diagnosis of chronic compartment syndrome before coming to our attention, delaying the proper diagnosis of PAES.

Non-invasive imaging modalities associated with a meticulous clinical examination, such as dorsiflexion and plantarflexion, causing popliteal artery compression and distal pulse disappearance, usually lead to an accurate diagnosis. In our series, all patients underwent active maneuvers such as dorsiflexion and plantarflexion, which can compress the popliteal artery and obliterate the blood flow during the examination. These maneuvers are imperative for the diagnosis of PAES in the early stage [16].

Thus, duplex ultrasonography represents a helpful tool to diagnose PEAS, especially when associated with provocative maneuvers obtained with calf muscle contraction [17,18].

However, to accurately determine the extent of the entrapment, multiple images are frequently necessary.

CT and MRA exams could define the diagnosis, showing in most patients the abnormalities on the popliteal artery and the surrounding tissues.

In our experience, invasive techniques such as the conventional angiography were used in almost all cases and represent an essential tool for patients who require endovascular treatment alongside surgical treatment. In fact, catheter-based local arterial thrombolysis is reported in case of complete artery occlusion, especially in an acute setting [19].

In our series, all patients underwent at least three examinations. In our series, the sensitivity to the detection of PAES was 96.2% for DSA with dynamic maneuvers, 54.5% (12 limbs) for MRA, and 51.7% (29 limbs) for CTA. Stearns at al. reported a sensitivity of 88% for MRA, reflecting a very high sensitivity of detection [18]. DSA, on the other hand, is more invasive but necessary, aiding in establishing a diagnosis and detecting arterial abnormalities with a 100% sensitivity [20].

The literature does not have a consensus with regard to the importance of DSA; in fact, some authors [21,22,23] considered catheter-directed angiography an invasive procedure, not more diagnostic than CTA and MRI, in recent times.

However, in our opinion, DSA associated with active maneuvers is the most diagnostic tool in our armamentarium, and it can be associated with preoperative fibrinolysis with Urokinase, performed to restore distal outflow, as reported in 4 cases in our experience.

Sinha et al. in a literature review of 26 PAES studies reported a bilateral involvement in around 40% of cases. In our series, most patients present a bilateral PAES, and even in case of asymptomatic contralateral popliteal entrapment discovered by radiological findings, to prevent definitive arterial damage or complications, surgical treatment is recommended.

Concerning surgical treatment, different surgical methods have been described, aiming for popliteal entrapment release, establishing a normal anatomy, and restoring a normal arterial flow. In most cases, in fact, surgical correction requires myotomy or resection of the aberrant musculotendinous structures. In addition, bypass is required in case of popliteal artery degeneration, with stenosis or occlusion, and post-stenotic dilatation or aneurysm formation.

With regard to the surgical technique, our practice recommends the posterior approach because the relationship between the musculotendinous tissues and the popliteal artery, as well as other anatomical features of the popliteal fossa, may generally be better observed using the posterior approach than the medial one.

Thus, in our experience, the preferred surgical approach was the posterior: 61 limbs (78.2%) received the posterior approach, whereas 17 (21.7%) the medial, especially in those cases where an extended surgical revascularization was necessary. As reported in a large Japanese retrospective multicenter study, the posterior approach was the main surgical option. Nevertheless, due to the low number of cases receiving the medial approach, a comparison between the two approaches is not reported [24].

Autologous venous substitution remains the best conduit to use, as reported in the literature [25], and vascular reconstruction with venous bypass is the most frequent procedure described in the literature [26]. In our experience, autologous venous bypass grafting was achieved in 18 limbs (23.1%), as only 1 limb received polytetrafluoroethylene (PTFE) graft due to the unavailability of a venous conduit.

Some authors suggest, as an adjunct to MTS, the endovascular therapy with thrombolysis to restore distal outflow. In our series, 4 limbs with poor outflow underwent preoperative thrombolysis with Urokinase and then MTS.

There was a clear difference in terms of arterial patency using arterial reconstruction or myotomy alone. This aspect could be justified not only because of the popliteal artery quality deterioration, but also because of the poor run-off below the knee arteries, caused by distal embolization from arterial or aneurysmal thrombosis.

Only a small number of studies provided a 5-year follow-up with a positive outcome following the surgical therapy (84–92%).

In our series, the rates for freedom from TLR and 15-year primary patency for the surgical treatment of PAES were 92.4% and 98%, respectively. Those patients treated with MTS had a 15-year patency rate of 98%, whereas those treated with reconstruction had a 71% patency rate at a 15-year follow-up.

Long-term patency was superior when musculotendinous sectioning was performed without vascular reconstruction.

## 5. Conclusions

Long-term results of surgical treatment for PAES seem to be very satisfying, and myotomy with or arterial reconstruction using venous bypass leads to good patency at 15 years of mid follow-up. The key to the success of the PAES syndrome treatment remains to be the exact and prompt diagnosis to avoid delays and irreversible arterial damage. 

## Figures and Tables

**Figure 1 diagnostics-14-01302-f001:**
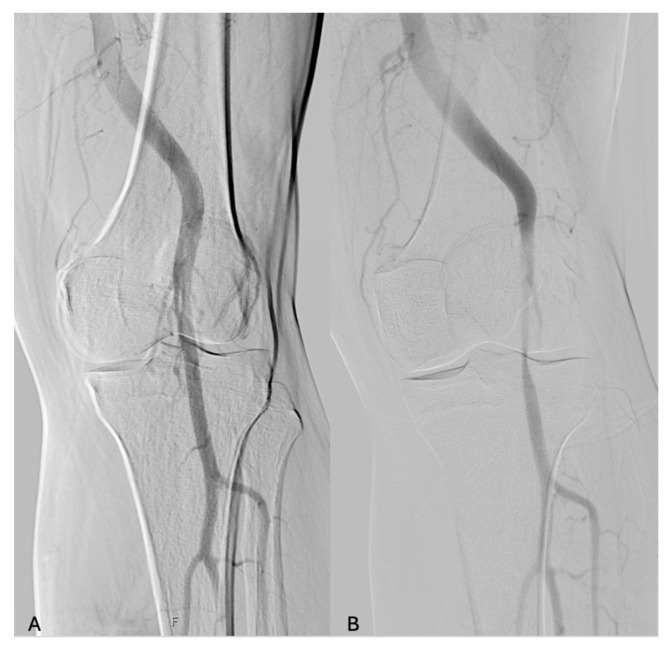
Digital subtraction angiography in a patient affected by PAES with popliteal artery stenosis after dynamic maneuvers. (**A**) neutral position; (**B**) dynamic maneuvers of dorsiflexion of the feet.

**Figure 2 diagnostics-14-01302-f002:**
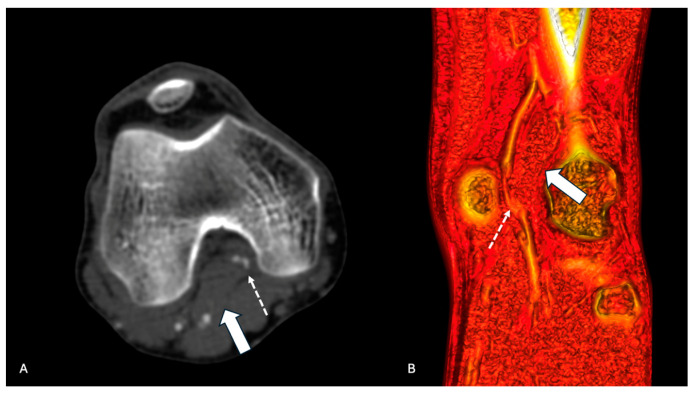
CTA of left inferior limb in a patient with PAES. (**A**) Coronal section of CTA showing anomalous insertion of the medial head of gastrocnemius (arrow) and occluded popliteal artery (dotted arrow). (**B**) Sagittal volume rendering showing anomalous insertion of the medial head of gastrocnemius (arrow) and occluded popliteal artery (dotted arrow).

**Table 1 diagnostics-14-01302-t001:** PVE Forum classification of popliteal entrapment syndrome.

PVE Forum Classification of Compressing Structures Causing Popliteal Entrapment
Type I: Popliteal artery running medially to the medial head of gastrocnemius
Type II: Medial head of gastrocnemius laterally attached
Type III: Accessory slip of gastrocnemius
Type IV: Popliteal artery passing below popliteal muscle
Type V: Primary venous entrapment
Type VI: Variants
Type F: Functional entrapment

**Table 2 diagnostics-14-01302-t002:** Demographic characteristics of patients with popliteal artery entrapment syndrome.

Total Patients (*n*)	47
Total Limbs (*n*)	78
Bilateral (*n*, %)	31 (39.7)
– Right	37 (47.4)
– Left	41 (52.5)
Type of PAES	
– Type I	0
– Type II	31 (39.7)
– Type III	23 (29.4)
– Type IV	0
– Type V	0
– Type VI	24 (30.7)
Median age (years)	34 (range, 14–62)
Sex (*n*, %)	
– Female	10 (22.7)
– Male	34 (77.2)
Athletes (*n*, %)	50 (64)
Comorbidities (%)	
– Smoking	41 (52.5)
– Diabetes Mellitus	0
– COPD	0
– CRF	0

**Table 3 diagnostics-14-01302-t003:** Preoperative clinical symptoms and type of imaging used for diagnosis.

Symptom classification * (*n*, %)	
– Class 0	4 (5.1)
– Class 1	32 (41)
– Class 2	30 (38.4)
– Class 3	10 (12.8)
– Class 4	0
– Class 5	2 (2.5)
Preoperative imaging (*n*, %)	
– Duplex ultrasound	78 (100)
– DSA	72 (97.4)
– CTA	56 (71.8)
– MRA	22 (28.2)

* Symptom classification: class 0: asymptomatic; class 1: pain, paresthesia, cold feet after physical activity; class 2: claudication (after >100 m); class 3: claudication (<100 m); class 4: rest pain; class 5: tissue loss (ulcer/necrosis). DSA: digital subtraction angiography; CTA: computed tomography angiography; MRA: magnetic resonance angiography.

**Table 4 diagnostics-14-01302-t004:** Types of surgical treatment and long-term patency rate.

Surgical treatment (*n*, %)	
– MTS *	58 (74.3)
– MTS + PTA *	2 (2.5)
– Autologous venous bypass	18 (23.1)
PTFE* bypass	1 (1.3)
Popliteal-popliteal	11
Femoro-popliteal	5
Femoro-anterior tibial artery	1
Femoro-posterior tibial artery	2
15-year patency rate (%, *p*-value)	
– MTS	98%
– Autologous venous bypass grafting	71%, *p* < 0.001

* MTS: musculotendinous section; PTA: percutaneous trans luminal angioplasty; PTFE: polytetrafluoroethylene.

## Data Availability

Data are not available due to privacy restrictions.

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
