# Peer review of "Long-Term Results of Surgical Treatment for Popliteal Artery Entrapment Syndrome"

_diagnostics, 2024, doi:10.3390/diagnostics14121302_

Round 1
Reviewer 1 Report
Comments and Suggestions for Authors
The manuscript “Long-term results of surgical treatment for popliteal artery entrapment syndrome” by Mansour is a research article in which the authors described their experience of treatment of popliteal artery entrapment syndrome (PAES) during a median 15-years period. Thus, patients treated for PAES in their institution from 1979 to 2024 were included, and preoperative, intraoperative and post-operative data were analyzed. The authors treated 47 patients with a total of 78 limbs, to all which duplex ultrasound with active maneuvers was performed. Angiography was performed in almost all patients while computed tomography angiography in 71.8% and magnetic resonance angiography in 28.2%. In regard to surgical treatment, musculotendinous section was performed in 76,9%, autologous venous bypass was achieved in 23.1%. Freedom from target lesion revascularization and fifteen-years primary patency were 92.4% and 98%, respectively. Thus, the authors concluded that long-term results of surgical treatment for PAES seem to be very satisfying. In addition, myotomy with or without arterial reconstruction using venous bypass can lead to good patency at 15 years of follow-up. In general, this article is critical in this field and contains essential findings. However, I have one comment before this manuscript is accepted for publication.
1. Tables should appear before figures.
Author Response
Dear Editors and Reviewers,
We’d like to express our profound gratitude for the comprehensive and insightful feedback provided on our manuscript,Long-term results of surgical treatment for popliteal artery entrapment syndrome. We deeply appreciate the time and expertise devoted by the reviewers in scrutinizing our submission. We have carefully reviewed each comment and have collaboratively worked on the revisions, ensuring that our responses address the suggestions and concerns raised effectively.
In the following sections, we detail our responses to each specific point of feedback, indicating the amendments made to the manuscript. Our goal has been to elevate the quality of our work, ensuring that it meets the high standards of Diagnostics and contributes significantly to the fields of vascular surgery, with a focus on cases, innovations, and techniques.
- Tables should appear before figures. [Thank you for your feedback on the manuscript. We appreciate your clarity and your comments on our manuscript. In response, we have modified as you have suggested]
Reviewer 2 Report
Comments and Suggestions for Authors
The research presented by the authors is very interesting and of practical value. A large series of PAES is presented in this study. However, I have questions and comments:
1. Page 1 line 26 What is meant by “target lesion revascularization”? redo revascularization?
2. Page 2 line 62 This sentence is unclear. The popliteal artery is a continuation of the superficial femoral artery.
3. Page 2 line 64 Replace “posterior tibial artery with tibioperoneal trunk.
4. Page 6 lines 224-229 All 78 limbs were surgically treated. The authors note that musculotendineous section was performed in 60 patients. In the following paragraph, the number of musculotendineos section in total do not equal 60 (31+5+4+2=42).
In addition, did the rest of the patients (18), who also had PAES not need to mesculotendineous section?
5. Page 6 line 244-246. Occluded grafts venous or prosthetic?
6. Page 9 line 317. Add a reference number to the Settembre’s report.
7. It would be interesting and informative for the readers the opinion of the authors (in the discussion) about differential diagnosis with an adventitial cystic disease and PAES
8. Reference 15 not complete (pp 434-438)
Author Response
Dear Reviewer,
We’d like to express our profound gratitude for the comprehensive and insightful feedback provided on our manuscript,Long-term results of surgical treatment for popliteal artery entrapment syndrome. We deeply appreciate the time and expertise devoted by the reviewers in scrutinizing our submission. We have carefully reviewed each comment and have collaboratively worked on the revisions, ensuring that our responses address the suggestions and concerns raised effectively.
In the following sections, we detail our responses to each specific point of feedback, indicating the amendments made to the manuscript. Our goal has been to elevate the quality of our work, ensuring that it meets the high standards of Diagnostics and contributes significantly to the fields of vascular surgery, with a focus on cases, innovations, and techniques.
- Page 1 line 26 What is meant by “target lesion revascularization”? redo revascularization?. [Thank you for your feedback on the manuscript. We appreciate your clarity. In response, we have revised the section to improve readability. We have clarified in the manuscript text, as follows:
“Freedom from target lesion revascularization, meaning that no significant stenosis or occlusion during follow up required revascularization” (Line 26-27)]
- Page 2 line 62 This sentence is unclear. The popliteal artery is a continuation of the superficial femoral artery. [Thank you for your nitpicking suggestion. Following your kind advice, we have revised the section to improve readability.We have clarified in the manuscript text, as follows:
“The proximal part of the popliteal artery originates as a continuation of the superficial femoral artery, which originates from the fusion of the ischiatic artery and the femoral artery” (line 64-65)]
- Page 2 line 64 Replace “posterior tibial artery with tibioperoneal trunk. [Thank you for your valuable suggestion to provide a more precise sentence. We have clarified in the manuscript text, as follows:
“The union of the tibialis anterior and tibioperoneal trunk forms the distal part”(line 66)]
- Page 6 lines 224-229. All 78 limbs were surgically treated. The authors note that musculotendinous section was performed in 60 patients. In the following paragraph, the number of musculotendinous sections in total do not equal 60 (31+5+4+2=42). In addition, did the rest of the patients (18), who also had PAES not need to musculotendinous section? [Thank you for raising this critical missed point. We missed to describe the others 18 patients that underwent different musculotendinous section. On the other hand, 18 patients affected by PAES did not need musculotendinous section and a simple release was done. Thus, we have revised the text, and clarified in the manuscript, as follows:
“10 limbs had an additional 3rd head of gastrocnemius; 6 limbs had a soleus sling and 2 limbs a fibrous band” (line 245;249-250)
“In the remaining 18 limbs, MTS was not performed, and a release of the musculotendinous structures was performed.” (line 245-247)]
- Page 6 line 244-246. Occluded grafts venous or prosthetic? [Thank you for raising this missed point. We have revised the text and clarified in the manuscript, as follows: “Both were venous graft” (line 266)]
- Page 9 line 317.Add a reference number to the Settembre’s report. [Thank you again for this oversight. We have revised the bibliography and correct it.]
- It would be interesting and informative for the readers the opinion of the authors (in the discussion) about differential diagnosis with an adventitial cystic disease (CAD) and PAES. [Thank you for your nitpicking suggestion. Following your kind advice, we have carefully expanded the section of our manuscript. This now includes a further description of differential diagnosis between Cystic Adventitial Disease and Popliteal artery entrapment syndrome. These additions aim to provide a clearer picture of two similar clinical scenario but characterized by different pathology. We believe these enhancements will greatly contribute to the depth and clinical relevance. We have revised the manuscript, as follows:
“CAD is a rare condition, accounting about 0.1% of vascular diseases. It is characterized by the development of a cystic mass in the sub-adventitial layer of the vessel. The pathognomonic sign of this condition is the visualization of a thin, echogenic line separating the lumen of the vessel and the cyst, with the narrowed lumen presenting with an ultrasonic scimitar sign on DUS imaging. In most cases, the neurovascular evaluation is normal. Passive flexion of the knee, however, always result in reduced distal pulses. This differs from PAES, where active plantar flexion or passive dorsiflexion maneuvers contracting the gastrocnemius muscle, reduce pedal pulses. Consequently, it is possible that a physical examination and diagnostic tests will not be able to differentiate between CAD and PAES; therefore, surgical planning should account for the possibility of encountering either condition.” (line 371-381)]
- Reference 15 not complete (pp 434-438). [Thank you for this oversight. We have corrected the reference which became number 16, as we added the reference number 15 for CAD (Smith et al.)]
Round 2
Reviewer 2 Report
Comments and Suggestions for Authors
I have no more comments or question.